# Using Pruning-Based YOLOv3 Deep Learning Algorithm for Accurate Detection of Sheep Face

**DOI:** 10.3390/ani12111465

**Published:** 2022-06-05

**Authors:** Shuang Song, Tonghai Liu, Hai Wang, Bagen Hasi, Chuangchuang Yuan, Fangyu Gao, Hongxiao Shi

**Affiliations:** 1College of Computer and Information Engineering, Tianjin Agricultural University, Tianjin 300384, China; shuang_song0809@163.com (S.S.); yuanchuangchuanga@163.com (C.Y.); gaofyu1@163.com (F.G.); 2Institute of Grassland Research, Chinese Academy of Agricultural Sciences, Hohhot 010010, China; wanghai@caas.cn (H.W.); hasibagen@caas.cn (B.H.)

**Keywords:** YOLOv3, sheep face recognition, K-means clustering, model compression

## Abstract

**Simple Summary:**

The identification of individual animals is an important step in the history of precision breeding. It has a great role in both breeding and genetic management. The continuous development of computer vision and deep learning technologies provides new possibilities for the establishment of accurate breeding models. This helps to achieve high productivity and precise management in precision agriculture. Here, we demonstrate that sheep faces can be recognized based on the YOLOv3 target detection network. A model compression method based on K-means clustering algorithm and combined with channel pruning and layer pruning is applied to individual sheep identification. In addition, the results show that the proposed non-contact sheep face recognition method can identify sheep quickly and accurately.

**Abstract:**

Accurate identification of sheep is important for achieving precise animal management and welfare farming in large farms. In this study, a sheep face detection method based on YOLOv3 model pruning is proposed, abbreviated as YOLOv3-P in the text. The method is used to identify sheep in pastures, reduce stress and achieve welfare farming. Specifically, in this study, we chose to collect Sunit sheep face images from a certain pasture in Xilin Gol League Sunit Right Banner, Inner Mongolia, and used YOLOv3, YOLOv4, Faster R-CNN, SSD and other classical target recognition algorithms to train and compare the recognition results, respectively. Ultimately, the choice was made to optimize YOLOv3. The mAP was increased from 95.3% to 96.4% by clustering the anchor frames in YOLOv3 using the sheep face dataset. The mAP of the compressed model was also increased from 96.4% to 97.2%. The model size was also reduced to 1/4 times the size of the original model. In addition, we restructured the original dataset and performed a 10-fold cross-validation experiment with a value of 96.84% for mAP. The results show that clustering the anchor boxes and compressing the model using this dataset is an effective method for identifying sheep. The method is characterized by low memory requirement, high-recognition accuracy and fast recognition speed, which can accurately identify sheep and has important applications in precision animal management and welfare farming.

## 1. Introduction

With the rapid development of farming intensification, scale and intelligence, the management quality and welfare healthier farming requirements of sheep farming are increasing, and sheep identification is becoming more and more important to prevent diseases and improve sheep growth. How to quickly and efficiently complete the individual identification of sheep has become an urgent problem. With the improvement of living standard, people’s demand for mutton products and goat milk products is increasing, while the quality requirements are also getting higher and higher. Therefore, large-scale farming to improve the productivity and production level of the meat and dairy industry is an effective way to increase farmers’ income, ensure food safety, improve the ability to prevent and control epidemics, and achieve the coordinated development of animal husbandry and the environment. Sheep identification is the basis of intelligent farm management, and currently common techniques for individual identification, such as hot iron tagging [1], freeze tagging [2], and ear notching [3], are available. These methods are prone to cause serious physical damage to animals [4]. Electronic identification, using methods like Radio Frequency Identification (RFID) tags, are widely used in the field of individual livestock identification [5]. However, this method is costly and prone to problems such as unreadability or fraud. To solve the above problems, experts have started working on new approaches by using deep learning methods. There has been an increase in the use of deep learning algorithms in the animal industry for different types of applications [6]. These applications include the biometric identification [7], activity monitoring [8], body condition scoring [9] and behavior analysis [10]. Recently, utilizing biometric traits instead of traditional methods for identifying individual animals has gained attention, due to the development of Convolutional Neural Networks (CNNs) [11,12,13]. To apply deep learning techniques to biometric recognition, it is necessary to acquire image information of specific body parts. Currently, a sheep retina [14], cattle body pattern [15] and the face of the pig have been used. However, obtaining clear data on the retina or body parts is very difficult [16]. Therefore, this paper proposes a contactless sheep face recognition method based on the improved YOLOv3. The contactless identification method-based biometric features was uniqueness. This method is not easily forged or lost, making contactless identification methods easier to use, more reliable and more accurate. Non-contact animal identification methods mainly include nasal print, iris recognition, etc. Compared with the contact identification method (such as ear incision, hot iron branding, freezing hit number, etc.), the application range is wide, the identification distance requirement is low, the operation is also relatively simple. Animal image data can be obtained by using portable photo devices such as cell phones and cameras, and the acquisition cost is low, saving a lot of human resources. The non-contact identification method does not require human–animal contact, which maximizes the avoidance of harm to animals and facilitates their growth and development. Biometric-based identification methods are mostly used in the identification of animals. The feature extraction method using Gabor filter has been attempted by Tharawat et al. Support vector machine classifiers with different kernels (Gaussian, polynomial, linear) [17] were compared. The results show that the classifier-based on Gaussian kernel is able to achieve 99.5% accuracy in the problem of nose-print recognition. However, the nasal pattern acquisition process is more difficult and the captured images contain much redundant information, so the method is not suitable for large-scale animal identification [18]. The iris of animals contains spots, filaments, crowns, stripes and other shape features, the combination of which does not change once the animal is born and is unique, so it can be one of the essential features for individual identification. Several studies have investigated the iris features of cattle using a two-dimensional complex wavelet transform feature approach. The iris images of cattle were captured and experimented with using a contactless handheld device, and the recognition accuracy reached 98.33% [19]. However, due to the large size of the iris capture device, the cost is high and the reliability of the recognition may be reduced due to image distortion caused by the lens. Therefore, it is not possible to extend the use of this method on a large scale. Simon et al. found that the human eye has a unique vascular pattern and that the retinal vessels are as unique as fingerprints. The retina is also present in all animals and is thus considered to be a unique individual biological characteristic. Rusk et al. [20] used retinal images of captured cattle and sheep to compare the images. The results showed that the recognition rate reached 96.2%. However, if the retina is damaged, the recognition effect will be greatly reduced, or even impossible.

The face is the most direct information about the external characteristics of an individual animal. Facial recognition is promising since it contains many significant features, e.g., eyes, nose, and other biological characteristics [21,22]. Due to the variability and uniqueness of facial features, the face of animals can become the identifier of individual recognition. Currently, the facial recognition technology for human has become increasingly sophisticated, but less research has been done on animals. Some institutions and researchers have been working on the research of animal facial recognition through unremitting efforts and continuous attempts. The migration of facial recognition technology to animals has been applied with good results. It provides a new idea for the study of facial recognition of livestock and poultry. Corkery et al. performed recognition of sheep faces and thus identified individual sheep. The overall analysis was performed using independent component techniques and then evaluated using a pre-classifier, with recognition rates of 95.3–96% [23]. However, fewer sheep face images were studied in this method, and the performance of the algorithm was not evaluated on a large sheep face dataset. Yang obtained more accurate results by analyzing the faces of sheep and introducing triple interpolation features in cascade pose regression [24]. However, changes in head posture or severe occlusion during the experiment can also lead to recognition failure. In 2018, Hansen et al. [13] used an improved convolutional neural network to extract features from pig face images and identify ten pigs. Due to the inconspicuous color and five senses of individual pigs, the overall environment is more complex and there are more other disturbing factors. This resulted in a low identification rate of individual pigs. Ma et al. proposed a Faster-RCNN neural network model based on the Soft-NMS algorithm. The method enables real-time detection and localization of sheep under complex breeding conditions. It improves the accuracy of recognition while ensuring the speed of detection. This model was able to detect sheep with 95.32% accuracy and mark target locations in real time, providing an effective database for sheep behavior studies [25]. Almog Hitelman et al. applied Faster R-CNN to localize sheep faces in images, providing the detected faces as input to seven different classification models. The best performance was obtained using the ResNet50V2 model with ArcFace loss function, and the application of migratory learning methods resulted in a final recognition accuracy of 97% [26]. However, in most studies, the generation of high levels of noise and large amounts of data due to different light sources and quality of captured images can also pose a challenge to the recognition process [27]. The sheep to be identified are in an environment with complex backgrounds, and the position and pose of the sheep face can also affect the recognition effect. The above method demonstrates that it is feasible to apply animal facial images to individual animal identification. In addition, there is still much room for improvement in data acquisition, data processing, recognition algorithms, and the accuracy of recognition.

In order to solve the above problems, this paper collected images of sheep faces in natural environment and produced a dataset. In addition, the sheep face images were data-enhanced before training to solve the overfitting problem caused by the small amount of data. The anchor boxes in YOLOv3 are clustered for the dataset of this study to improve the accuracy of model identification. Then, the model is compressed to save resources and make it easy to deploy, so that the final model is only 1/4 of the initial model. The accuracy is also higher than the initial model. The model is effectively simplified while ensuring the detection accuracy. By establishing a dataset of images of sheep and realizing the training and testing of the model. The results show that the method can greatly improve the detection and identification efficiency, which is expected to make China’s livestock industry go further toward achieving intelligent development, help ranchers optimize sheep management and create economic benefits for the sheep farming industry.

## 2. Materials and Methods

### 2.1. Research Objects

In this experiment, we used Sunit sheep, a sheep breed located in Xilin Gol League, Inner Mongolia, as the subject. The face of this breed of sheep is mostly black and white, so it is also called “panda sheep”. Due to the timid nature of sheep, people cannot get close enough, making it more difficult to photograph sheep faces data. Prior to the experiment, the sheep were herded from their living area into a closed environment around them and left to calm down.

### 2.2. Experimental Setup

The experiment was completed by the experimenter holding the camera equipment, as shown in Figure 1. The enclosed environment in which the sheep were placed was rectangular, 5 m long and 3 m wide. Each sheep was taken in turn to another enclosed environment and filmed individually when they were calm. The deep learning model was implemented in a Windows environment by writing code based on the pytorch framework. Training was performed on a PC with NVIDA GeForce GTX3090 Ti.

### 2.3. Data Collection

The dataset used in this study was obtained from the Sunit Right Banner Desert Grassland Research Base, Institute of Grassland Research, Chinese Academy of Agricultural Sciences. The collection date was 29 June 2021, within the experimental plots, and sheep face data were collected for each sheep in turn. The dataset collected video data from 20 adult Sunit sheep. The videos were recorded at a frame rate of 30 frames per second, a frame width of 1920 pixels, and a frame height of 1080 pixels. In order to make the collected dataset with a certain complexity, different lighting and different angles are used for the shooting. Therefore, each sheep’s face was captured from different perspectives. Finally, a total of 20 sheep were captured, with 1–3 min of video captured for each sheep. In this study, the sheep number was the ear tag number of the sheep.

### 2.4. Dataset Creation and Preprocessing

Follow the steps below to create the sheep face dataset: First, a video of each sheep’s face was manually marked with the sheep’s ear tag number. Next, python code was used to convert each video file into an image sequence. Finally, 1958 sheep face images were obtained by manually removing the images with blurred sheep faces and the images without sheep faces for data cleaning.

To facilitate image annotation and avoid the problem of overfitting of training results, structural similarity index measurement (SSIM) [28] was performed to remove overly similar images. SSIM measures luminance and contrast. A final comparison of luminance, contrast and structure was performed to analyze the similarity between the two input images. In addition, the problem of uneven dataset acquisition due to the different characteristics of individual sheep. For example, some sheep are active by nature, so the collected data are sparse, and some prefer to be quiet, resulting in dense data collection. We can use the data enhancement method to solve the problem of uneven data distribution between different classes. Flipping, rotating and cropping some of the sheep face images can improve the generalization ability of the model and solve the problem of uneven distribution of data classes. In other words, this can prevent the model from not learning enough features or learning a large number of individual features to degrade the model performance. Some examples of the enhancement are shown in Figure 2. A total of 2400 images of sheep face were collated, 120 images per sheep; of these, 1600 were used to train the sheep face recognition model, 400 were used to perform model validation, and the remaining 400 were used to evaluate the final performance of the model. Using the labeling tool for target border annotation, generate XML formatted annotation file after annotation. The file content includes the folder name, image name, image path, image size, sheep only category name, and pixel coordinates of the labeled box, etc. Figure 3 shows the labeling results.

## 3. Sheep Face Recognition Based on YOLOv3-P

The aim of our research was to recognize sheep faces through intelligent methods. Inspired by face recognition, we tried to apply deep learning methods to sheep face recognition. In this section, we introduce the proposed K-means clustering and model compression methods.

### 3.1. Overview of the Network Framework

With the rapid development of deep learning, target detection networks have shown excellent performance in part recognition. Inspired by deep learning-based face recognition methods, we improve the YOLOv3 target detection network to achieve facial recognition of sheep. In this work, we propose the YOLOv3-P network to optimize the network in terms of recognition accuracy and model size, respectively.

### 3.2. Sheep Face Detection Based on YOLOv3

The YOLOv3 [29] network model is one of the object detection methods that evolved from YOLO [30] and YOLOv2 [31]. The YOLOv3 neural network was derived from the Darknet53 and consists of 53 convolutional layers, which uses skip connections network inspired from ResNet. It has shown the state-of-the-art accuracy but with fewer floating points and better speed. YOLOv3 uses residual skip connection to achieve accurate recognition of small targets through upsampling and cascading. It is similar to the way the feature pyramid network performs detection at three different scales. In this way, YOLOv3 can detect targets of any size. After feeding an image into the YOLOv3 network, the detected information, i.e., bbox coordinates and objectness scores, as well as category scores, are output from the three detection layers. The predictions of the three detection layers are combined and processed using a non-maximum suppression method, and then the final detection results are determined. Unlike the Faster-R-CNN [32], YOLOv3 is a single-stage detector that formulates the detection problem as a regression problem. YOLOv3 can detect multiple objects by reasoning at once, so the detection speed is very fast. In addition, by applying a multistage detection method, it is able to compensate for the low accuracy of YOLO and YOLOv2. The YOLOv3 framework is shown in Figure 4. The input image is divided into n × n grids. If the center of a target falls within a grid, that grid is responsible for detecting that target. The YOLOv3 network achieves target detection by clustering the COCO dataset as candidate frames. The COCO dataset contains 20 categories of targets, ranging in size from large buses and bicycles to small cats and birds. If the parameters of the anchor frame are directly used for the training of the sheep face image dataset, the detection results will be poor. Therefore, K-means algorithm is used to recalculate the size of the anchor frame based on the sheep face image dataset. A comparison of the clustering results is shown in Figure 5. The red pentagrams in the figure are the anchor frame scales after clustering, while the green pentagrams represent the anchor frame scales before clustering. Set the anchor boxes of the collected sheep face dataset after clustering to (41, 74), (56, 104), (71, 119), (79, 146), (99, 172), (107, 63), (119, 220), (156, 280), (206, 120). Each feature map is assigned 3 different sizes of anchor boxes for bounding box prediction.

### 3.3. Compress the Model by Pruning

YOLOv3 has a greater advantage in sheep face recognition accuracy compared with other target detection networks, but it is still difficult to meet the requirements of most embedded devices in terms of model size. Therefore, model compression is needed to reduce the model parameters and make it easy to deploy. The methods of model compression are divided into pruning [33,34], quantization [35,36], low-rank decomposition [37,38] and knowledge distillation [39,40]. Pruning has become a hot research topic in academia and industry because of its ease of implementation and excellent results. Fan et al. simplify and speed up the detection by using channel pruning and layer pruning methods for the YOLOv4 network model. This reduces the trimmed YOLOv4 model size and inference time by 241.24 MB and 10.82 ms, respectively, The mean average precision (mAP) also improved from 91.82% to 93.74% [41]. A pruning algorithm combining channel and layer pruning for model compression was proposed by Lv et al. The results show a 97% reduction in the size of the original model, while the speed of reasoning increased by a factor of 1.7. At the same time, there is almost no loss of precision [42]. Shao et al. also proposed a method combining network pruning and YOLOv3 for aerial IR pedestrian detection; Smooth-L1 regularization is introduced on the channel scale factor; This speeds up reasoning while ensuring detection accuracy. Compared with the original YOLOv3, the model volume is compressed by 228.7 MB, and the model AP is reduced by only 1.7% [43]. The above method can effectively simplify the model while ensuring image quality and detection accuracy, providing a reference for the development of portable mobile terminals. In this article, inspired by web thinning, it proves to be an effective pruning method. So we choose a pruning algorithm that combines channel and layer pruning to compress the network model.

#### 3.3.1. Channel Pruning

Channel pruning is a coarse-grained but effective method. More importantly, the pruned model can be easily implemented without the need for dedicated hardware or software. Channel pruning for deep networks is essentially based on the correspondence between feature channels and convolutional kernels. Crop off a feature channel and the corresponding convolution kernel to achieve model compression. Suppose the i-th convolutional layer of the network is x_i_, and the dimension of the input feature map of this convolutional layer is ($h_{i},w_{i},n_{i}$), where $n_{i}$ is the number of input channels, and $h_{i},w_{i}$ denote the height and width of the input feature map respectively. The convolution layer transforms the input feature map $x_{i}\in R^{n_{i}+h_{i}+w_{i}}$ into the output feature map $x_{i+1}\in R^{n_{i+1}\times h_{i+1}\times w_{i+1}}$ by applying $n_{i+1}$ 3D filters $F_{i,j}\in R^{n_{i}\times k\times k}$ on $n_{i}$ channels, and is used as the input feature map for the next convolutional layer. Each of these filters consists of $n_{i}$ 2D convolution kernels $K\in R^{k\times k}$, and all filters together form the matrix $F_{i}\in R^{n_{i}\times n_{i+1}\times k\times k}$. As shown in Figure 6, suppose the current convolutional layer is computed as $n_{i+1}n_{i}k^{2}h_{i+1}w_{i+1}$. When the filter $F_{i,j}$ is pruned, its corresponding feature map $x_{i+1,j}$ will be deleted. This reduces $n_{i}k^{2}h_{i+1}w_{i+1}$ times of computation. At the same time, the filter corresponding to $x_{i+1,j}$ in the next layer will also be pruned, which again additionally reduces the computation of $n_{i+1}k^{2}h_{i+2}w_{i+2}$. That is, the m filters in layer i are pruned, and the computational cost of layers i and j can be reduced at the same time, which finally achieves the purpose of compressing the deep convolutional network.

#### 3.3.2. Layer Pruning

The process of layer pruning for the target detection network is similar to that of channel pruning. Layer pruning cannot depend only on the principle that the number of convolutional kernels in adjacent convolutional layers is preserved equally. Because one of the convolutional layers at each end of the shortcut structure in YOLOv3 is cut off, it not only affects itself, but also has a serious impact on the whole shortcut structure, which can cause the network to fail to perform operations. Consequently, one of the convolutional layers at each end of the shortcut structure in YOLOv3 is cut off; it not only affects itself, but also has a serious impact on the whole shortcut structure, which can cause the network to fail to perform operations. When a shortcut structure is determined to be unimportant, the entire shortcut structure is cut off. Due to the small amount of operations in the shallow network, instead of pruning the first shortcut structure, we start from the second shortcut structure and the retrograde layer pruning operation. To begin with, the model is trained sparsely, and the stored BN layer γ parameters are traversed in the second shortcut structure, starting with a 3 × 3 convolution kernel. In addition, the L_1_ parameters of γ of each BN layer are compared and the L_1_ parameters of γ of the 2nd convolutional kernel BN layer in these 22 shortcut structures are ranked. The second BN layer s for the i-th shortcut structure can be expressed as: $L_{i}=\overset{N_{c}}{\underset{s=1}{\sum }}\left|\gamma_{s}\right|$, where: $N_{c}$ denotes the number of channels. $L_{i}$ is the L_l_ norm of the second BN layer y parameter of the i-th Shortcut, which indicates the magnitude of the importance of the Shortcut structure. Next, take the smaller L_n_ (n is the number of shortcut structures to be subtracted), the shortcut structure is cut out accordingly, where each shortcut structure includes two convolutional layers (1 × 1 convolutional layer and 3 × 3 convolutional layer), hereby the corresponding 2 × L_n_ convolutional layers are clipped. Figure 7 depicts the layers before and after pruning.

#### 3.3.3. Combination of Layer Pruning and Channel Pruning

Channel pruning has significantly reduced the model parameters and computation, and reduced the model’s resource consumption, and layer pruning can further reduce the computation effort and improve the model inference speed. By combining layer pruning and channel pruning, the depth and width of the model can be compressed. In a sense, small model exploration for different datasets is achieved. Therefore, this paper proposes to perform channel pruning and layer pruning by performing both on the convolutional layer. Find more compact and efficient convolutional layer channel configurations to reduce training parameters and speed up training. For this purpose, both channel pruning and layer pruning were applied in YOLOv3-K-means to obtain the new model. The model pruning process is shown in Figure 8. The optimal pruning ratio is found in 0.05 steps incrementally until the model mAP reaches the threshold value of effective compression, and the model pruning ends. We finally chose a cut channel ratio of 0.5. In addition to that, we also cut out 12 shortcuts, that is, 36 layers. After the pruning is over, it does not mean the end of the model pruning work, as the weight of the clipped sparse model part is set to 0. If it is used directly for detection tasks, its model accuracy will be greatly affected. Consequently, the model needs to be further trained on the training set to recover the model performance, a process called iterative tuning.

### 3.4. Experimental Evaluation Index

To evaluate the performance of the model, the model was compared with several common deep learning-based object detection models in terms of accuracy, size of training weights, and inference speed. For the object detection model, accuracy and recall are two relatively contradictory metrics. As the recall rate increases, the accuracy rate decreases and vice versa. The two metrics need to be traded off for different situations. Adjustment is made by model detection confidence thresholds. In order to evaluate the two metrics together, the F1 score metric was introduced. It is calculated as in Equation (3). In addition to this, the average accuracy value of all categories, mAP, is used to evaluate the accuracy of the model, as shown in Equation (4).
(1)$$P=\frac{\mathrm{TP}}{\mathrm{TP}+\mathrm{FP}}$$
(2)$$R=\frac{\mathrm{TP}}{\mathrm{TP}+\mathrm{FN}}$$
(3)$$F1-\mathrm{score}=\frac{2*P*R}{P+R}$$
(4)$$\mathrm{mAP}=\frac{1}{M}\overset{M}{\underset{k=1}{\sum }}\mathrm{AP}\left(k\right)$$
where TP is the number of true positives, FP is the number of false positives, P is the precision, R is recall, and F1-score is a measure of test accuracy, which combines precision P and recall R to calculate the score. AP(k) refers to the average precision of the kth category, whose value is the area under the P(precision)–R(recall) curve; M represents the total number of categories, and mAP is the average of all categories precision.

## 4. Experimental Results

### 4.1. Result Analysis

#### 4.1.1. Experiment and Analysis

Model training and testing were performed using the Windows operating system. The test framework is PyTorch 1.3.1 (Facebook AI Research, New York, NY, USA) (CPU is Intel(R) Xeon(R) Silver 4214R, 128GB RAM, NVIDA GeForce GTX3090 Ti) and the parallel computing framework is CUDA (10.1, NVIDIA, Santa Clara, CA, USA). The hyperparameters of the model were set to 16 samples per batch with an initial learning rate of 0.001. The model saves weights every 10 iterations for a total of 300 iterations. As the training proceeds, the loss function continues to decrease. By the time the iteration reaches 300, the loss function is no longer decreasing.

In this paper, some experiments are conducted to compare the improved models with different backbone networks of Faster-RCNN, SSD, YOLOv3 and YOLOv4. The performance of each model was validated on the same dataset. Retain the model generated after 300 training sessions are completed, and the profile corresponding to the model. Test the mAP, Precision, Recall, Speed and model parameters of the above network model with the same test code with the same parameters and record the test results. Figure 9 represents the changes of each evaluation criterion of the proposed models during the training process. It can be seen that the accuracy and loss values of the model gradually level off as the number of iterations increases. Figure 10 shows the recognition effect of the improved model for sheep faces.

#### 4.1.2. Comparison of Different Networks

In order to select a more suitable model for improvement, models based on different backbone networks were compared. Table 1 shows the detection results of different models on the same sheep face dataset. Compared with the CSP Darknet53-based YOLOv4 model, the Darknet53-based YOLOv3 model was 4.15% and 1.20% higher in mAP and F1-score, respectively. Compared with Faster-RCNN, a two-stage detection model based on the Resnet network, in mAP and F1-score were 5.10% and 4.70% higher, respectively. YOLOv3 as a single-stage detection algorithm also outperforms the two-stage detection algorithm Faster-RCNN in terms of speed. Lastly, YOLOv3 is compared with the SSD model based on the VGG19 network. Both the mAP and F1-score of YOLOv3 are slightly lower than those of the SSD model. However, there are many more candidate frames in the SSD model than YOLOv3, resulting in a much slower detection speed than YOLOv3. After comparing all aspects, we finally chose the YOLOv3 model for optimization. In addition, we performed K-means clustering of anchor frames in YOLOv3 based on the sheep face dataset. Compared with the initial model, the mAP improved by 1.1%.

#### 4.1.3. Comparative Analysis of Different Pruning Strategies

In order to evaluate the performance of the proposed model, the YOLOv3 model based on the clustering of sheep face dataset is pruned to different degrees, which includes channel pruning, layer pruning and simultaneous pruning of channels and layers. The results of the final fine-tuning of the model are compared. Table 2 shows the detection results with different pruning levels. Compared with the model after clustering based on the sheep face dataset, its pruned model has a slight improvement in both mAP and detection speed. The model size has also been reduced from 235 MB to 61 MB, making it easier to deploy. Compared with the original YOLOv3 model, the proposed model for sheep face dataset clustering based on YOLOv3 with pruned layers and pruned channels reduces the number of convolutional layers and the number of convolutional layers and the number of network channels by a certain amount. The mAP, accuracy, recall and F1-score all improved by 1.90%, 7%, 1.80% and 5.10%., respectively. The detection speed was also improved from 9.7 ms to 8.7 ms, while the number of parameters has been significantly reduced. To make the results more convincing, we mixed the training and test sets and trained the model 10 times using a 10-fold cross-validation method. The results are shown in Figure 11. The histogram represents the mean value of 10 experiments, and the error bars represent the maximum and minimum values of 10 experiments with mAP value of 96.84%.

### 4.2. Discussion

It is worth noting that the sheep face data collected using cameras is relatively large compared to data collected by other sensors. If the data need Internet for transmission, it may be a challenge. This is especially true for ranches in areas with poor Internet connectivity. For this reason, our vision is to propose that sheep face recognition be performed using edge computing, thus improving response time and saving bandwidth. The processed results are transmitted instead of the original video to a centralized server via cable or a local area network on a 4G network. Such an infrastructure is now available for modern farms for monitoring purposes. The extracted information will be stored and correlated in the server. In addition to this, we are able to apply sheep face recognition to a sheep weight prediction system to further realize non-contact sheep weight prediction, thus avoiding stress reactions in the sheep that could affect their growth and development.

In this paper, we present a method for identifying sheep based on facial information using a deep learning target detection model. In order to ensure that the method is accepted in practical applications, in the short-term future plans, we will focus on (a) Significantly increasing the number of sheep tested to verify the reliability of sheep face recognition, (b) Development of a specialized sheep face data collection system with edge computing capabilities, (c) Optimize shooting methods to avoid blurred, obscured images, etc. as much as possible, as well as, (d) The model is further compressed by attempting parameter quantization, network distillation, network decomposition, and compact network design to obtain a lightweight model. Moreover, maintaining accuracy while compressing is also an issue to be considered. In the long-term plan for the future, we will develop more individual sheep monitoring functions driven by target detection algorithms and integrate them together with edge computing.

## 5. Conclusions

In this paper, we propose a deep learning-based YOLOv3 network for contactless identification of sheep in pastures. This study further extends the idea of individual sheep identification through image analysis and considers animal welfare issues. Our proposed method uses YOLOv3 as the base network and modifies the anchor boxes in the YOLOv3 network model. The IoU function is used as the distance metric of the K-means algorithm. The K-means clustering algorithm is used to cluster the sheep face dataset to generate new anchors, replacing the original anchors in YOLOv3. It makes the network more applicable to the sheep face dataset and improves the accuracy of the network model in recognizing sheep faces. In order to make the proposed network model easy to deploy, the clustered network model is compressed. At the same time, the cut channel and cut layer operations were performed. The test results show that, under normal conditions, the F1-score and mAP of the model reached 93.30% and 97.20%, respectively. It significantly demonstrates the effectiveness of our strategy on the test dataset. It facilitates the daily management of the farm and also provides a good technical support for the smart sheep farm to achieve welfare farming. The proposed model is capable of inferring images of size 416 pixels at 8.7 ms on GPU, which has the potential to be applied to embedded devices. However, there are still some shortcomings in our method, such as the small sample size of the experiment. In the future, we will try to test the ability to identify large numbers of sheep by applying our model in a real ranch and consider the application of sheep face recognition to sheep weight prediction system.

## Figures and Tables

**Figure 1 animals-12-01465-f001:**
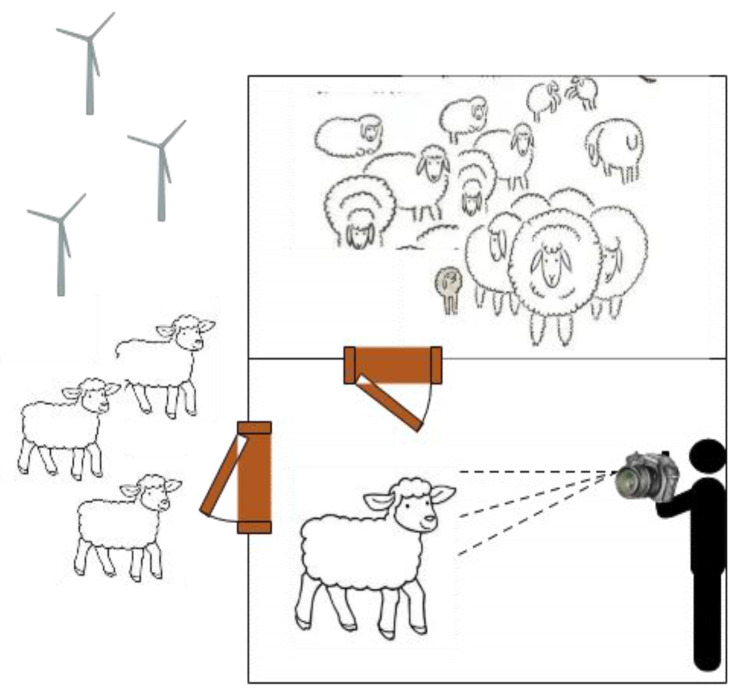
The layout of the experimental scene.

**Figure 2 animals-12-01465-f002:**
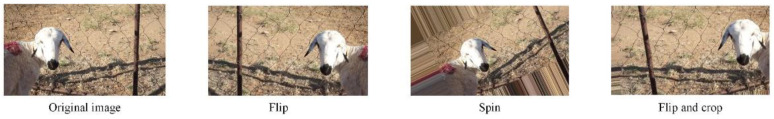
Data Augmentation.

**Figure 3 animals-12-01465-f003:**
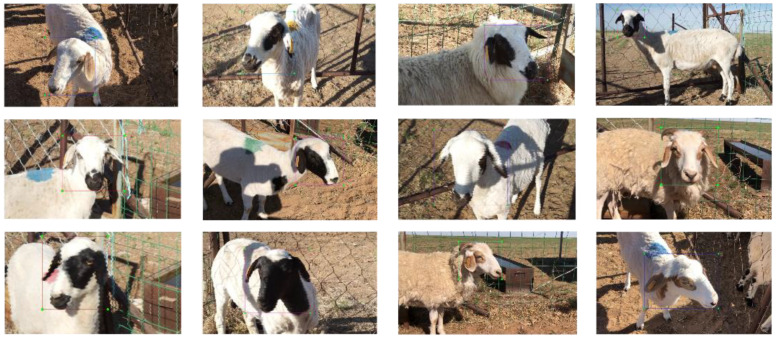
Labeling results of dataset.

**Figure 4 animals-12-01465-f004:**
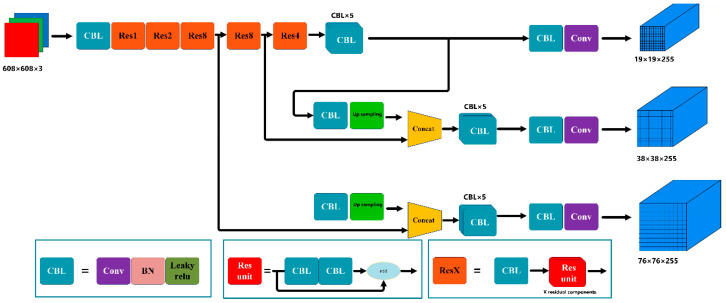
YOLOv3 structure. CBL is the smallest component of the YOLOv3 network architecture, which consists of Conv (convolution) + BN + Leaky relu; Res unit is the residual component; ResX, X stands for number, there are Res1, Res2, …, Res8, etc., which consists of a CBL and N residual components.

**Figure 5 animals-12-01465-f005:**
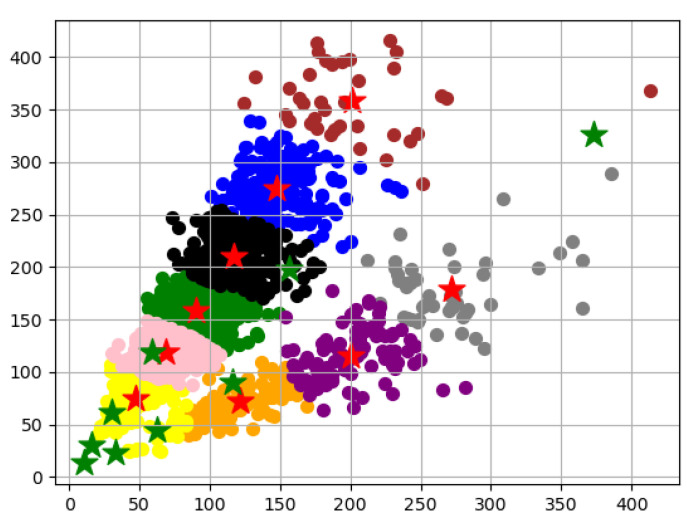
Clustering results of anchor boxes. The red pentagrams in the figure are the anchor frame scales after clustering, while the green pentagrams represent the anchor frame scales before clustering. Set the anchor boxes of the collected sheep face dataset after clustering to (41, 74), (56, 104), (71, 119), (79, 146), (99, 172), (107, 63), (119, 220), (156, 280), (206, 120).

**Figure 6 animals-12-01465-f006:**
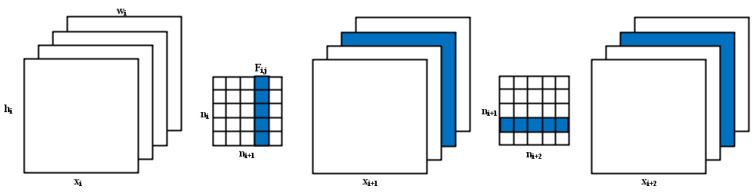
Channel pruning principle. $x_{i}$ denotes the i-th convolutional layer of the network; $h_{i},w_{i}$ denote the height and width of the input feature map respectively; $n_{i}$ is the number of input channels of this convolutional layer.

**Figure 7 animals-12-01465-f007:**
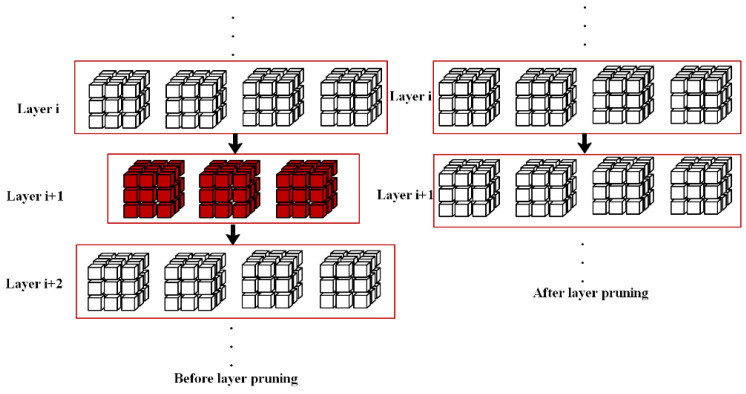
Before and after layer pruning.

**Figure 8 animals-12-01465-f008:**
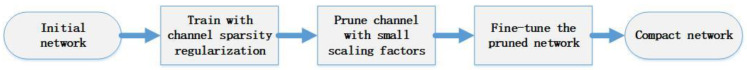
Flowchart of network slimming procedure.

**Figure 9 animals-12-01465-f009:**
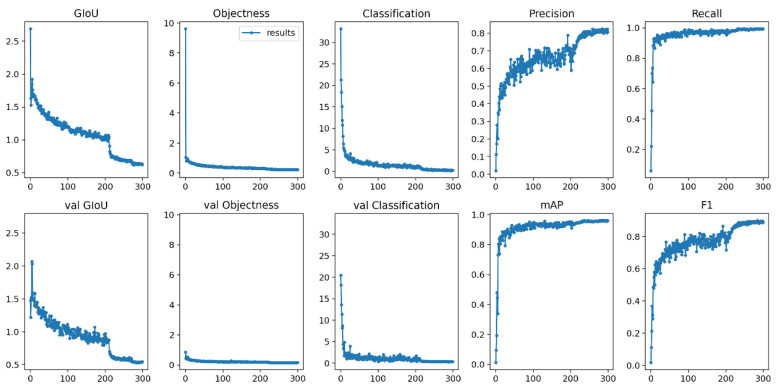
Changes in evaluation criteria during the training process.

**Figure 10 animals-12-01465-f010:**
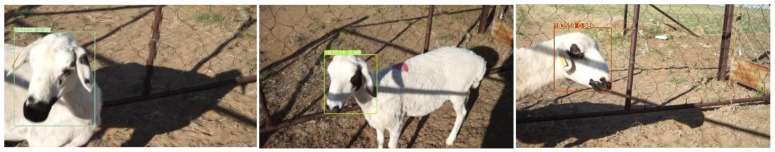
Recognition result.

**Figure 11 animals-12-01465-f011:**
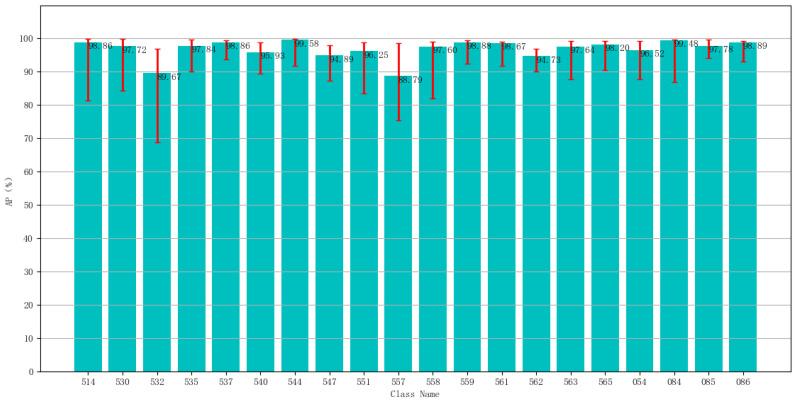
10-Fold cross-validation results of our model.

**Table 1 animals-12-01465-t001:** Comparison of different network detection results.

Model	mAP	Precision	Recall	F1-Score	Parameters
Faster R-CNN	90.20%	80.63%	90%	84.00%	108 MB
SSD	98.73%	96.85%	96.25%	96.35%	100 MB
YOLOv3	95.30%	82.90%	95.70%	88.70%	235 MB
YOLOv4	91.15%	88.70%	88.00%	87.50%	244 MB

**Table 2 animals-12-01465-t002:** Comparison of the results of different levels of pruning.

Model	mAP	Precision	Recall	F1-Score	Parameters	Speed
Prune_channel	96.80%	89.50%	97.00%	92.80%	69.9 MB	8.9 ms
Prune_layer	95.70%	89.50%	95.70%	91.90%	132 MB	9.2 ms
Prune_channel_layer	97.20%	89.90%	97.50%	93.30%	61.5 MB	8.7 ms

## Data Availability

The dataset can be obtained by contacting the corresponding author.

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
