# Peer review of "Using Pruning-Based YOLOv3 Deep Learning Algorithm for Accurate Detection of Sheep Face"

_animals, 2022, doi:10.3390/ani12111465_

Round 1

Reviewer 1 Report

  • The citations should include more recent references (5 years)
  • The author should include relevant existing work on channel pruning/layer pruning in a deep network since this approach is not new
  • Authors should provide an illustration of how clustering of anchor boxes affects the initial anchor boxes selected
  • The number of samples (20) is quite small to conclude that the method can actually recognize sheep faces, or sheep can be recognized by their faces, rather than other physical features.
  • Nevertheless, this is an interesting research 

Reviewer 2 Report

The user proposed a deep learning method of sheep face using YOLOv3. K-means algorithm was proposed but I did not see any result of  K-means visualization graph i.e. scatter graph to show the effectiveness of the method. K-means used centroid based method that group classification by cluster, therefore the user should show the visualization result. The authors should add the graph.

In overall, the result showed very good result of precision and recall.

Reviewer 3 Report

It is difficult to find a contribution of this study to readers.

Authors apply known techniques(K-means for anchor selection and pruning for model compression) to sheep face detection.

Furthermore, this study is not for sheep face recognition but for sheep face detection.

Many studies have been reported for animal detection by using deep learning-based object detectors.

If this study is combined with face recognition, it will be interesting to readers.

Round 2

Reviewer 3 Report

This manuscript deals with face detection, not face recognition.

(So, the title of this manuscript should be changed.)

Furthermore, K-means clustering is included in YOLO.

(So, this part should not be emphasized.)

Finally, pruning is a well-known technique for model compression.

(So, it is difficult to find any originality in this manuscript.)

Author Response

请参阅附件。

Round 3

Reviewer 3 Report

The manuscript was well revised.